# Modelling the Rheology of Olive Paste for Oil Extraction Plant Automation: Effects of the Crushing Process on the Rheology of Olive Pastes

**DOI:** 10.3390/foods12112218

**Published:** 2023-05-31

**Authors:** Claudio Perone, Filippo Catalano, Alessandro Leone, Antonio Berardi, Antonia Tamborrino

**Affiliations:** 1Department of Soil, Plant and Food Science (DISSPA), University of Bari Aldo Moro, Via Amendola 165/a, 70126 Bari, Italy; claudio.perone@uniba.it (C.P.); alessandro.leone@uniba.it (A.L.); antonio.berardi@uniba.it (A.B.); 2CTS s.r.l.—Spin-Off, Department of Agriculture, Environment and Food, University of Molise, Via Francesco De Sanctis, 86100 Campobasso, Italy

**Keywords:** olive paste viscosity model, extraction yield, solids packing factor, paste rheology

## Abstract

In extra virgin olive oil production, it is essential to obtain a well-prepared olive paste which allows not only the extraction of the oil drops from the olives, but also the achievement of a high-quality oil while maintaining high yields. This work addresses the problem of determining the effect of three crushing machines on the viscosity of the olive paste: a hammer crusher, a disk crusher and a de-stoner were tested. The tests were repeated on both the paste leaving each machine and the paste to which water was added; this was done with the main aim of considering the different dilutions of the paste while entering the decanter. A power law and the Zhang and Evans model were used to analyse the rheological behaviour of the paste. The experimental results allow validation of the two models with a high (more than 0.9) coefficient of determination between experimental and numerical data. The results also show that the pastes obtained with the two classic crushing methods (hammers and disks) are almost identical, with a packing factor of about 17.9% and 18.6%, respectively. Conversely, the paste obtained with the de-stoner entails higher viscosity values and a smaller solid packing factor, of about 2.8%. At 30% dilution with water, the volume of the solid concentration dropped to about 11.6% for the hammer and disc crushers, while for the de-stoner it only reached 1.8%. This behaviour is also reflected in the evaluation of yields, which were 6% lower with the de-stoner. No significant differences regarding the legal parameters of oil quality were found using the three different crushing systems. Finally, this paper establishes some fundamental pillars in the research for an optimal model for identifying the rheological behaviour of the paste as a function of the crusher used. Indeed, since there is an increasing need for automation in the oil extraction process, these models can be of great help in optimizing this process.

## 1. Introduction

Olive paste preparation is an essential stage within the olive oil extraction process. The crushing of olives is not only a simple physical process used to break the fruit’s tissues and release the oil drops contained in the vegetal cell vacuoles, but also a critical step that affects the quality of the final virgin olive oil (VOO) produced. The final characteristics of the oil can be modified according to the crushing machine employed [1,2,3], as well as by varying the size of the diameter of the outlet holes of its external grid [4]. Additionally, the crusher speed of the tools plays an important role in olive oil quality, as faster crushing significantly increases chlorophyll and total biophenol content [5]. The same results were obtained by [6]. They found that extraction efficiency, chlorophyll content, total phenol content, 3,4-DHPEA-EDA, and p-HPEA-EDA increase with faster hammer mill rotor speed and smaller screen size. The effects of the mechanical components of the crushers on pits and oil droplet fractionation were studied in [7]. The authors assert that the analysis of the experimental data shows that different crushing machines produce different pit and oil drop fragmentation, and the trends confirm that increased pit fragmentation corresponds to increased droplet fragmentation. Regarding yield, [8] stated that no statistically significant differences were observed in terms of oil yield when different metal crusher devices were used. This was also confirmed by [9], who found hammer and knife crushers did not show significant variations in the extractability of the final olive oil. However, using a total de-stoner in olive paste preparation led to a decrease in extraction yield, although it produced a higher-quality oil [10,11]. To preserve the high quality of the product without affecting the yield, it is possible to use partial de-stoners, reducing the solids content to about 40–50% [12].

After the crushing phase, the olive paste undergoes a further operation of conditioning in order to generate the optimal conditions for the final solid–liquid separation (decanter centrifuge) and liquid–liquid separation (vertical centrifuge). The way in which the olive paste is processed after crushing also greatly influences yield and quality. Generally, the conditioning happens through malaxation, mainly aimed at oil coalescence and enzymatic activation [13]. In an effort to improve oil quality and extractability, several studies were conducted recently on novel technologies to pre-treat and/or condition the olive paste. Some of these are heat exchangers [14,15], microwaves [16], ultrasounds [17,18,19] and pulsed electric fields [20,21,22,23]. All these processes, with malaxation first, induce important modifications in the rheology of the olive paste, mainly due to stress–strain effects and temperature effects [14,24,25].

After conditioning, the olive paste is moved into a decanter centrifuge for solid–liquid separation, which probably represents the most important step in raising the oil yield. Ref. [26] pointed out that one of the main physical parameters influencing the solid–liquid separation in a decanter centrifuge is the olive paste viscosity. In fact, it affects the velocity gradients and consequently the motion of suspended solid particles that must be removed. To this aim it is important to identify the right pre-treatment of the crushed paste in order to optimize the operation of the decanter centrifuge. Ref. [27] introduced a malaxating machine prototype equipped with a torque-monitoring system to investigate the rheological parameters of olive paste (in-line measures) and to try to identify when the paste is in the best condition to progress to the next solid–liquid separation. This is done in the attempt to achieve higher yields and higher-quality products based on the input raw material.

On the one hand, everything mentioned so far is expensive to implement, as it requires a great deal of experimental activity; on the other hand, it can be partially obtained using suitable software frameworks that allow the simulation of at least some of the activities described above [28,29]. Modelling real objects in a virtual world allows not only the historicization of data (and obviously their visualization), but above all the simulation of possible future evolutions and imagined responses to extreme external stimuli in carrying out sensitivity tests. This allows an engineer to test, analyse and optimize the setting virtually before any actual changeover is conducted. The inter-networking of the different objects which are embedded with sensors, actuators and models enables the devices to communicate with each other as necessary. It also helps with decision making and allows real-time responses. To this end, it is necessary to develop suitable numerical models, obviously validated through experimental tests, which can be incorporated into the various frameworks which can be found mainly through energy analysis of the building microclimate [30] of the food industry [31,32,33,34], and this subject is generally well covered by scientific papers, but not on numerical modelling of the olive oil extraction process. In fact, there are only a few studies, concerning only some of the machines used: [31,35] for the decanter modelling and [36] for the malaxer modelling, as well as some studies regarding the modelling of olive paste rheology as a function of some processing parameters [26,37,38]. However, to date, the literature is lacking in recent papers specifically concerning the modelling of olive paste after the crushing phase (see, for example, an old paper by [39]). Therefore, in this study, an investigation is conducted on the effect on olive paste rheology of different crushing machines at different dilutions of water. As reported in [38] and confirmed by [26], olive paste is a shear thinning non-Newtonian fluid well represented by the power law model. The power law model was therefore employed to identify paste consistency and flow behaviour index. These parameters were then used to derive a general model to identify paste behaviour at different dilutions with water. This work represents an important starting point to identify the rheological behaviour of paste as a function of the crushing method used and how the addition of water could affect its rheology. With a view to pushing more and more automation of the oil extraction process, these models represent key functions in optimizing the settings of the entire process [29], with particular attention to the subsequent phases of conditioning (temperature and time) and solid–liquid separation (bowl speed, differential speed, etc.). This can be achieved by equipping the plant with a model-based controller which, by predicting the viscosity of the incoming olive paste, determines the optimal dilution (sub-optimal when other parameters such as quality are added).

## 2. Materials and Methods

### 2.1. Extraction Process Line with Different Crushers

Figure 1 shows the experimental plant for the olive oil extraction process. It consists of a first stage in which the olives loaded into the hopper (H) were cleaned by a defoliation machine (D) (Mercuri Co., Ltd., Rosarno-RC, Italy) and a washing machine (W) (Santoro Co., Ltd., Ceglie Messapica-BR, Italy). Then, thanks to a manifold equipped with valves (MWV), the olives were delivered to the selected crusher: total de-stoner (TOT-DM) (Alfa Laval Co., Ltd., Lund, Sweden) with mass flow rate of 1.2 t h^−1^, hammer crusher (HC) (Alfa Laval Co., Ltd., Lund, Sweden) with mass flow rate of 3.7 t h^−1^ or disk crusher (DC) (Alfa Laval Co., Ltd., Lund, Sweden) with mass flow rate of 1.7 t h^−1^. Soon after, the olive paste was moved into the managing section thanks to a cavity pump (CP) made of 6 malaxing machines with a capacity of 750 L each (Alfa Laval Co., Ltd., Lund, Sweden). The conditioned paste was then subjected to separation in a 2-phase decanter centrifuge (DEC) (Alfa Laval Co., Ltd., Lund, Sweden) and the liquid phase obtained was further separated by means of 2 vertical centrifuges (VC) (Alfa Laval Co., Ltd., Lund, Sweden).

### 2.2. Rheological Behavior of Olive Paste

The olive paste samples taken at the output of each crusher in operation (HC, DC or TOT-DM) were analysed to identify their rheological behaviour. As reported in [26], olive paste is a non-Newtonian fluid and thus we must refer to apparent viscosity (*μ*), since it varies as a function of shear rate and shear stress. The power law model was utilized for the measurement of apparent viscosities at different shear rates:(1)$$\mu =m{\dot{\gamma }}^{n-1},$$
where $\dot{\gamma }$ (s^−1^) is the shear rate, m is the consistency index (Pa s*^n^*) and *n* is the flow behaviour index (dimensionless). The process described in Section 2.3 led to the identification of the consistency and flow behaviour index of the olive paste sampled. Since olive paste is a heterogeneous system made of water, oil and hard (pits) and soft (pulp) solids which are suspended in the fluid phase, the parameters extrapolated from the power law model were then employed in the Zhang and Evans model to evaluate the rheological behaviour of this dispersed system. The Zhang and Evans model [40], which is a generalized Quemada method [41] where the constant is zero, was chosen because it yields the best results on olive paste as assessed by [26], confirming that modelling the crushed olive paste as a fluid (water, oil and a small amount of the soft solids) with all other solids suspended in it is a good choice. Therefore, this model can be used to analyse how the rheological behaviour of olive paste varies with a solid load:(2)$$\mu_{r}={\left(\frac{\varphi_{max}-c\varphi }{\varphi_{max}-\varphi }\right)}^{2}={\left(\frac{1-c\frac{\varphi }{\varphi_{max}}}{1-\frac{\varphi }{\varphi_{max}}}\right)}^{2},$$
where $\mu_{r}=\mu /\mu_{l}$ ($\mu_{l}$ is the viscosity of the fluid in Pa s^n^) is the relative viscosity (dimensionless), $\varphi =V_{s}/\left(V_{t}\right)$ (volume of suspended solids in m^3^ on total volume in m^3^) is the solid volume fraction (dimensionless), $\varphi_{max}$ its maximum value (packing factor, no flow possible) and c is a constant to be determined experimentally (dimensionless).

In particular, the solid volume fraction (dimensionless), $\varphi_{max}$ and the viscosity of the fluid, $\mu_{l}$, correspond (see [26,41] for details) to the *x*-axis intercept and to the *y*-axis intercepts of the Quemada model respectively (graphs not shown); the constant of the model, *c*, was determined by the least square method between measured data and model data. The method follows the procedure of [26] to simplify model calculation without the need for particle size distribution determination or other more sophisticated techniques, such as SEM images. This helps the automation of the olive oil extraction plant as few variables are involved to determine the model, and they can even be carried out online according to the procedure shown in [27].

### 2.3. Experimental Plan

A homogeneous lot of olives of the Nocellara del Belice (*Olea europaea* L.) cultivar were divided into 12 sub-lots weighing 700 kg each for each crushing method. Each test was repeated four times, and the olive pastes were sampled at the outlet of the crusher in operation and placed into 1000 mL glass containers conditioned at 27 °C in a thermostatic bath. After the crushing phase, the paste was conditioned in the malaxer for 30′ at 27 ± 1 °C. The decanter operated without adding water.

For each test, the extracted oil was measured, and the oil extractability (E) was calculated according to [42]. During each test, samples of olive paste were taken at the outlet of the crusher to determine the content of water, solids and oil according to [43].

The olive paste samples were then diluted with the addition 10, 20 or 30% water (Table 1) to analyse the rheological behaviour at different solid concentrations. All the prepared samples were analysed with a Brookfield rotational viscometer (Brookfield Engineering Laboratories, Inc., Stoughton, MA, USA) equipped with interchangeable disc spindles, 2–7 (model RV/HA/HB; Brookfield DVII + Brookfield Engineering Laboratories). The apparent viscosities were measured using 600 mL of olive paste and recorded at 10 rotational speeds from 0.5 to 100 rpm, using the RV/HA/HB-4 spindle. Data processing was carried out by linear regression in a bi-logarithmic scale [26].

Since the flow behaviour index can be considered constant in each crushing condition at different dilutions, as we will explain in Section 3.2, the consistency index evaluated with the power law model was used instead of apparent viscosity to determine the constant *c* of Equation (2) (as reported in [26]. Based on the paste composition (Table 1) and relative densities, *φ* and *φ_max_* were also evaluated.

### 2.4. Legal Quality Parameters

Free acidity, peroxide value and spectrophotometric constants (K_232_, K_270_ and ΔK) of oils extracted were evaluated according to Regulation (EU) 2015/1830 (OJEC, 2015).

### 2.5. Statistical Analysis

One-way analysis of variance (ANOVA) and Tukey HSD were used to evaluate the significance of the extractability data. Statistical analysis was performed using the Statistica 6.0 software package.

## 3. Results

### 3.1. Olive Oil Extractability (E) and Quality

In Table 2, the olive oil extractability and quality for the three olive paste preparation systems used are shown.

The data show that using the hammer or disc crusher makes no significant difference in E, while using the de-stoning machine leads to a significant loss of E. The loss of extraction yield with the use of the de-stoning machine had already been highlighted in other scientific works; in fact, in [44], a yield loss of 1.6 kg of oil per 100 kg of processed olives was detected using olives of the Coratina variety. The significantly lower yield by using the de-stoning machine was also observed in [45]. The authors hypothesize that the yield loss is due to two main aspects related to the absence of stone fragments in the olive pastes, which (i) reduces the effect of breaking cell walls and vacuoles during the malaxing phase and (ii) causes a reduction in draining effects, making the separation of liquids and solids more difficult during the centrifugation of the pastes in the decanter. Finally, the authors assert that to limit the loss of oil, a reduction of the olive paste feed rate to the decanter, a better adjustment of the difference in revolutions between bowl and screw and an increase in malaxation time could be effective. The olive oil quality of the experimental trials carried out by using different crushing systems did not identify any significant alteration to the legal quality parameters of the olive oils when compared to the control samples (Table 2).

The values of acidity, peroxide index and spectrophotometric constants were in the legal limits of the category of EVOO, showing the high-quality characteristics of the final product.

### 3.2. Rheological Model

The results of the power law model were used to identify the consistency and flow behaviour indices, which represent the starting point for the rheological modelling of the dispersed system. In fact, thanks to these values, it was possible to evaluate the constant c of the Zhang and Evans models in each crusher configuration. The main results are shown in Table 3. The resulting models yield a numerical prediction of experimental values with a coefficient of determination (R^2^) greater than 0.9 (see Figure 2 and Figure 3) and with very small RMSE and P of the power law model (see Table 3), confirming that the power law model is the most suitable for the main aim of the paper, that is, to determine a complete model of olive paste to automate decision making for dilution before entering the decanter. It is possible to note that no significant difference was observed in flow behaviour index when the crushing method was varied. The same result was also observed in a previous study by [26]. This fact implies that in our assessments, we can use the consistency index instead of the apparent viscosity. The consistency index undergoes a high decrease as the percentage of dilution increases. In the case of the hammer crusher, we observed a decrease of about 80% passing through 0% to 30% water addition, while for the disc crusher and total de-stoning machine, respective decreases of about 77% and 81% were recorded.

Since the composition of the input product is almost the same, no significant differences were observed in the consistency of the fluid. Another important result is that the constant *c* can be considered with good approximation to be zero in all cases. This means that the general model of Zhang and Evans can be reduced to a Quemada model with little more approximation, and all calculations can be carried out by considering only the ratio $\varphi /\varphi_{max}$, allowing an important simplification that is very useful in plant automation. It is important to note that $\varphi_{max}$ in the case of TOT-DM is about 49% lower than in the cases of HC and DC. This is mainly due to higher colloidal interaction forces between particles when the pits were removed. In fact, in this case, soft solids and pulp tend to aggregate, encapsulating a great deal of solvent (liquid phase) and resulting in a higher viscosity.

### 3.3. Paste Rheology According to the Crushing Method

Figure 2 shows the comparison of the apparent viscosity of different crushing methods, taking the hammer crusher as a reference. Disc crushing and hammer crushing led to the same results in terms of apparent viscosity. In fact, the slope of the regression line is unitary, with a coefficient of correlation of about 0.99. In the case of total de-stoning machine, the slope is higher by about 16% compared to the hammer crusher (and hence also to the disc crusher).

The hard solid particles (pits) in the shear flow undergo rotation in the main direction of the flow, but since they have an irregular shape, they sweep out a higher volume compared to spherical particles (such as in emulsions), resulting in a higher effective volume fraction [46]. Indeed, as reported in Figure 3, the solid volume fractions of olive paste obtained by HC and DC, respectively, are quite similar and higher than those obtained by TOT-DM at each dilution. When TOT-DM was used as a crushing method, the volume solid fraction range is very narrow, with a “packing factor” ($\varphi_{max}$) of 2.8% (Table 3). At 0% dilution, φ is about 2.3% and at 30% dilution, the volume of the solid is only reduced to about 1.8%. This confirms that when the pits are totally removed, soft solids tend to aggregate, reducing the fluidity of the liquid phase and creating a colloidal system [27]. The relative viscosity at 0% dilution in the case of TOT-DM is higher by about 10% compared to HC and by about 21% compared to DC.

The rheological behaviour of olive paste obtained by means of HC and DC is extremely similar, especially with dilution increases. Only at 0% water dilution is the relative viscosity of the paste produced by HC higher, by about 12%. This is probably due to the particles’ size, since HC produces smaller particles compared to DC [47]. Smaller particles produce a higher viscosity because, at a given value of φ, they show a reduced interparticle distance and a higher effect of repulsive interactions, which for a small particle is higher than the attractive forces. Near the particles, the flow is disturbed because of a local increase of the velocity gradient, and with a higher φ the flow disturbance produced by the particles starts to overlap [46].

## 4. Conclusions

The crushing method has an enormous impact on the whole extraction process, since olive paste preparation affects the final characteristics of the oil and extraction yield. The way in which the olives are milled influences the rheology of the olive paste. The rheology is also influenced by the subsequent conditioning phase (malaxation, heating, etc.). To understand the flow behaviour of olive paste, three crushing methods were analysed and compared in this study: hammer crushing (HC), disc crushing (HC) and total destoning (TOT-DM). The olive pastes obtained were analysed by means of the power law model and then the Zhang and Evans model was used to explain their rheological behaviour at different dilutions of water. The power law model showed no significant difference in the flow behaviour index in all three cases. However, the consistency in the case of TOT-DM was higher compared to HC and DC, which were quite similar. Since the flow behaviour index is almost the same, consistency in place of apparent viscosity was used in the Zhang and Evans model to study the dispersion systems at different water dilutions. The packing facto, *φ_max_*, of the paste produced by TOT-DM was significantly lower (2.8%) compared to HC and DC (17.9% and 18.6% respectively). At 30% water dilution, the volume of suspended solids, φ, was reduced to only 1.8%. In the case of HC and DC, *φ* dropped to about 11.6%. This observed behaviour in TOT-DM paste is probably due to a behaviour similar to a colloidal system. As for extractability, no significant difference was observed between HC and DC, while TOT-DM was 6% lower.

The crushing system used did not significantly affect the legal oil quality parameters; moreover, all the oils obtained during the experimentation fell within the legal criteria of EVOO.

This study confirms the importance of correctly modelling the rheological behaviour of the pastes produced during crushing so as to be able to optimize the settings of the subsequent conditioning and extraction phases, which are considerably affected.

## Figures and Tables

**Figure 1 foods-12-02218-f001:**
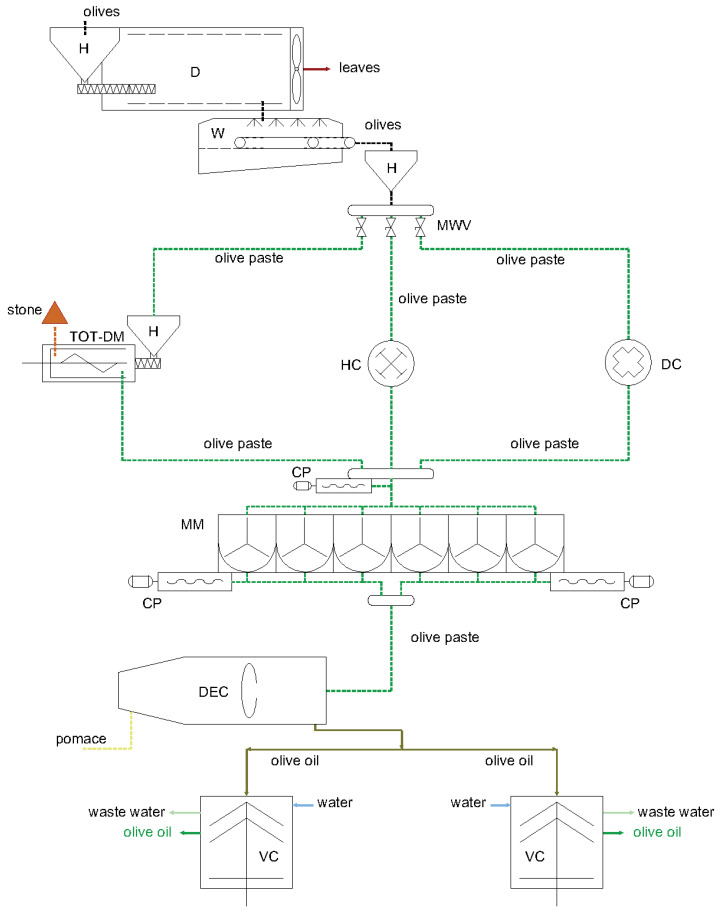
Layout of the experimental plant with hammer crusher (HC), disc crusher (DC) and total de-stoning machine (TOT-DM).

**Figure 2 foods-12-02218-f002:**
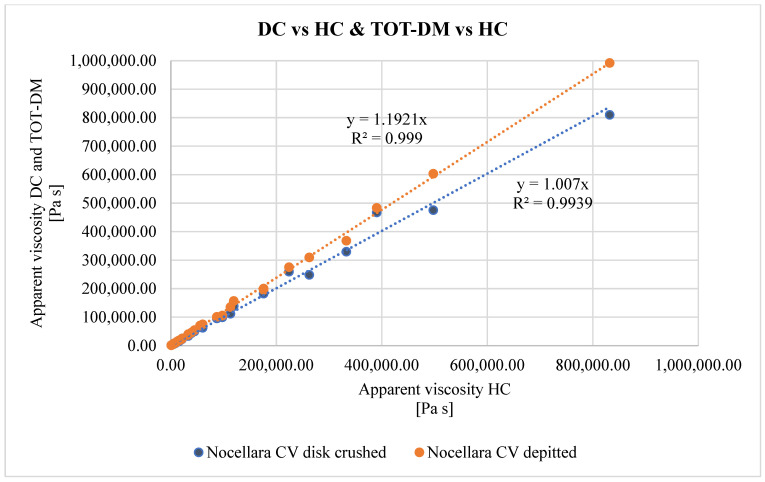
Comparison of apparent viscosity from different crushing methods.

**Figure 3 foods-12-02218-f003:**
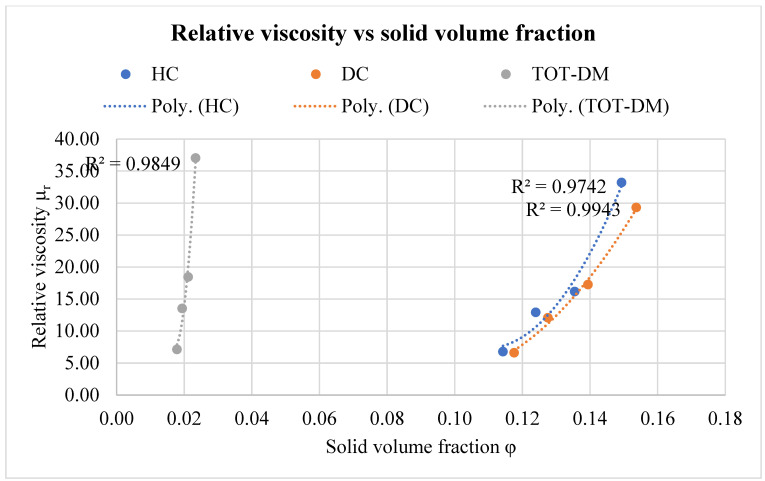
Relative viscosity as a function of solid volume fraction.

**Table 1 foods-12-02218-t001:** Experimental test identification of different crushing methods and water dilutions.

ID	Crushing Method	Dilution Water	Composition		
		(%)	Water (%)	Oil (%)	Solids (%)
HC 1	Hammer crusher	0	60.46	20.69	18.85
HC 2		10	64.05	18.80	17.13
HC 3		20	67.05	17.24	15.70
HC 4		30	69.58	15.91	14.50
DC 1	Disc crusher	0	60.2	20.54	19.26
DC 2		10	63.81	18.67	17.50
DC 3		20	66.83	17.11	16.05
DC 4		30	69.38	15.80	14.81
TOT-DM 1	Total de-stoning machine	0	69.20	23.95	6.85
TOT-DM 2		10	72.00	21.77	6.22
TOT-DM 3		20	74.33	19.95	5.70
TOT-DM 4		30	76.30	18.42	5.26

**Table 2 foods-12-02218-t002:** Olive oil extractability and legal parameters.

Crushing Method	E	Acidity	Peroxide Value	K_232_	K_270_	∆K
	(%)	(%)	(meq O_2_/Kg Oil)			
HC	88.45 ± 1.23 a	0.16 ± 0.02 a	4.0 ± 0.9 a	1.522 ± 0.016 a	0.113 ± 0.014 a	−0.003 ± 0.001 a
DC	86.17 ± 1.11 a	0.15 ± 0.01 a	3.8 ± 0.7 a	1.582 ± 0.015 a	0.110 ± 0.016 a	−0.003 ± 0.001 a
TOT-DM	80.30 ± 1.42 b	0.17 ± 0.02 a	4.1 ± 0.6 a	1.574 ± 0.016 a	0.114 ± 0.016 a	−0.003 ± 0.001 a

Data expressed as means of four replicates ± standard deviation. Different letters indicate statistically significant difference (*p* < 0.05).

**Table 3 foods-12-02218-t003:** Results of power law model and Zhang and Evans model.

ID	m	n			$m_{l}$	c	$\varphi_{max}$
	Pa s*^n^*	Mean [-]	RMSE	P [%]	Pa s*^n^*	[-]	[-]
HC 1	154,451	0.844 ± 0.0283 a	0.0245	2.403	4652	0.0300	0.179
HC 2	75,219						
HC 3	60,127						
HC 4	31,583						
DC 1	147,905	0.849 ± 0.0191 a	0.0165	1.636	5048	0.0493	0.186
DC 2	87,131						
DC 3	60,962						
DC 4	33,405						
TOT-DM 1	186,996	0.835 ± 0.0174 a	0.0151	1.461	5050	0.0430	0.0277
TOT-DM 2	93,173						
TOT-DM 3	68,279						
TOT-DM 4	35,981						

Data expressed as means ± standard deviation. Different letters indicate statistically significant difference (*p* < 0.05).

## Data Availability

Data is contained within the article.

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
