# Peer review of "Modelling the Rheology of Olive Paste for Oil Extraction Plant Automation: Effects of the Crushing Process on the Rheology of Olive Pastes"

_foods, 2023, doi:10.3390/foods12112218_

Round 1

Reviewer 1 Report

The crushing process is a very important step, as the preparation of olive paste affects the final quality properties of the oil and the oil yield in extraction. Since water dilution was made in olive paste in this study, the scientific soundness of the manuscript would be excellent if some physicochemical oil quality parameters were also studied along with phenolics apart from rheology. I believe this manuscript will discuss the issue in detail and contribute to the olive oil industry. An original, well-planned work on the subject. 

The publication year of some articles used as references is very old. There are newly published studies on some of them. Please replace the old ones with new ones.

Reference 1 is not assigned correctly, please correct it. In addition, all references should be revised according to author guidelines and DOI numbers added.

Author Response

The crushing process is a very important step, as the preparation of olive paste affects the final quality properties of the oil and the oil yield in extraction.

Since water dilution was made in olive paste in this study, the scientific soundness of the manuscript would be excellent if some physicochemical oil quality parameters were also studied along with phenolics apart from rheology. I believe this manuscript will discuss the issue in detail and contribute to the olive oil industry. An original, well-planned work on the subject. 

R.: Thank you for the suggestion. As pointed out by the reviewer, analyzing the quality of the oil would have made the manuscript even more complete. However, combining the study and modeling of the rheology of the olive paste with the qualitative aspects of the oil would have led to the writing of a manuscript that was too long with two different methodological and investigative approaches. We believe that each of these aspects is essential for understanding the effects of the machine on the matrix and deserve a dedicated treatment, and to this regard the study of the qualitative aspects has been planned in a subsequent scientific investigation.

The publication year of some articles used as references is very old. There are newly published studies on some of them. Please replace the old ones with new ones.

R.: Thank you for the suggestion. The rheology models used in this study were introduced many years ago, and they are still heavily used in more recent publications. However, their use for the analysis of olive paste rheology is lacking, and the few studies known to the authors are already cited in the paper.

Reference 1 is not assigned correctly, please correct it. In addition, all references should be revised according to author guidelines and DOI numbers added.

R.: Thank you for the comment, reference 1 has been corrected.

Reviewer 2 Report

The paper represents a very interesting study linking science and industry of olive processing, and is therefore, of great importance for the readers in the scientific, as well as in the industrial community. There are some issues which need to be corrected, mostly connected to some analyses which have not been done, but can influence the rheology of the paste, as well as some missing statistical data, which confirm the adequacy of the models. Please see comments below.

Please make a slight change in the first part of the title to: "Modeling the rheology of olive paste.....".

P1, L39: What is meant by "crusher speed"? Maybe the flow of material through the crusher or the rotating speed of the crushing elements? Please be more specific and rewrite this sentence.

P4, L132: Please list the crushers in this sentence to make the text easier to follow.

P5, L137-138: Please add the measuring units with the description of the symbols for the power law model.

P5, L149-152: Please add measuring units for each symbol.

Materials and methods: Particle size analysis (or at least microscope or SEM images of the paste) should be added to the manuscript, since it greatly influences the rheological behaviour.

P6, L185, Table 2: How were the results in Table 2 obtained? How many parellel measurements were done to calculate the mean shown in this table? Please add the valid N. Also, the row where DC is shown should present the E number using decimal points instead of comma.

P7, Table 3: Please add statistical data to table 3 (you discuss significant differences later in the text, so there should be significant differences shown in the table). Also, please add the R2 values for the developed models, as well as RMSE and P (%) error values in Table 3, since they confirm the adequacy of the developed models.

Author Response

The paper represents a very interesting study linking science and industry of olive processing, and is therefore, of great importance for the readers in the scientific, as well as in the industrial community. There are some issues which need to be corrected, mostly connected to some analyses which have not been done, but can influence the rheology of the paste, as well as some missing statistical data, which confirm the adequacy of the models. Please see comments below.

Please make a slight change in the first part of the title to: "Modeling the rheology of olive paste.....".

R.: Thank you for the suggestion, the title has been corrected.

P1, L39: What is meant by "crusher speed"? Maybe the flow of material through the crusher or the rotating speed of the crushing elements? Please be more specific and rewrite this sentence.

R.: Thank you for the comment. The speed was referred to the rotating speed of tools. It was better specified in the manuscript.

P4, L132: Please list the crushers in this sentence to make the text easier to follow.

R.: Thank you for the suggestion, the sentence has been corrected. The list of crushers has been also added to the caption of Figure 1.

P5, L137-138: Please add the measuring units with the description of the symbols for the power law model.

R.: Thank you for the suggestion, the unit of measurements have been added.

P5, L149-152: Please add measuring units for each symbol.

R.: Thank you for the suggestion, the unit of measurements have been added.

Materials and methods: Particle size analysis (or at least microscope or SEM images of the paste) should be added to the manuscript, since it greatly influences the rheological behaviour.

R.: Thank you for the observation. We know that, generally speaking, the rheological behaviour of a fluid with dispersed solids may depend on the particle size distribution but, in this case and according to Boncinelli et al. [26], we have used a greatly simplified model that uses only few parameters (only the constant c of the model is evaluated by interpolation) that can be measured also on line according to the procedure shown in [27]. Text was added to explain the above comment at the end of paragraph 2.2.

P6, L185, Table 2: How were the results in Table 2 obtained? How many parellel measurements were done to calculate the mean shown in this table? Please add the valid N. Also, the row where DC is shown should present the E number using decimal points instead of comma.

R.: Thank you for the observation. A clarification sentence has been added after Table 2, the number of replicates has been added and decimal point was used instead of comma

P7, Table 3: Please add statistical data to table 3 (you discuss significant differences later in the text, so there should be significant differences shown in the table). Also, please add the R2 values for the developed models, as well as RMSE and P (%) error values in Table 3, since they confirm the adequacy of the developed models.

R.: Thank you for the observation: RMSE and P (%) values are added in table 3 and discussed in the test.

Reviewer 3 Report

This article is interesting, and deserves to be published. However, it is unfortunate that the authors do not offer information about some quality parameters that could have been affected depending on the crusing method.

For example, in Table 2, it would be necessary to indicate in addition to the extractability, also the acidity, the oleic content, the peroxide index, and some diene indexes.

I suggest completing the analysis, and resubmitting the manuscript.

Author Response

This article is interesting, and deserves to be published. However, it is unfortunate that the authors do not offer information about some quality parameters that could have been affected depending on the crusing method.

For example, in Table 2, it would be necessary to indicate in addition to the extractability, also the acidity, the oleic content, the peroxide index, and some diene indexes.

I suggest completing the analysis, and resubmitting the manuscript.

R. Thank you for your suggestion. As reviewer 1 points out, although an analysis of the quality of the oil would have made the manuscript even more complete, a discussion of it would be desirable in a future manuscript. In fact, combining the study and modeling of the rheology of the olive paste with the qualitative aspects of the oil would have led to the drafting of a too long manuscript with two different methodological and investigative approaches. We believe that each of these aspects is essential for understanding the effects of the machine on the matrix and deserves a dedicated discussion. In this regard, the study of the qualitative aspects was foreseen in a subsequent scientific investigation.

Round 2

Reviewer 2 Report

All of the questions were addressed and answered adequately, so I do not have further comments. I suggest accept.

English is OK.

Author Response

We would like to thank the reviewer for the support.

Reviewer 3 Report

I disagree with the author's answer. Adding a small table or figure for the quality of the oil obtained by each crushing method would not have added excessive space to this manuscript, and would have allowed a better discussion of the suitability of each method. In fact, in its present form, this manuscript is not very long.

Author Response

We would like to thank the reviewer for the additional comments. The authors agreed with the comments and corrected thw text according to the comments.